# Critical Evaluation of Common Claims in Loop Quantum Cosmology

**Martin Bojowald** 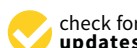

Institute for Gravitation and the Cosmos, the Pennsylvania State University, 104 Davey Lab, University Park, PA 16802, USA; bojowald@gravity.psu.edu

**Abstract:** A large number of models have been analyzed in loop quantum cosmology, using mainly minisuperspace constructions and perturbations. At the same time, general physics principles from effective field theory and covariance have often been ignored. A consistent introduction of these ingredients requires substantial modifications of existing scenarios. As a consequence, none of the broader claims made mainly by the Ashtekar school—such as the genericness of bounces with astonishingly semiclassical dynamics, robustness with respect to quantization ambiguities, the realization of covariance, and the relevance of certain technical results for potential observations—hold up to scrutiny. Several useful lessons for a sustainable version of quantum cosmology can be drawn from this evaluation.

**Keywords:** loop quantum cosmology; quantum corrections; effective field theory; covariance

## 1. Introduction

Loop quantum cosmology is based on potential modifications of space–time structure such as an underlying discreteness of space. Additional, less obvious quantum space–time effects may be implied by the resulting modifications of canonical constraints generating hypersurface deformations in space-time. A general analysis is, therefore, expected to be challenging and rather counter-intuitive. Nevertheless, several very detailed and optimistic claims have been made about the deep quantum behavior of the theory, in particular about how the big-bang singularity may be resolved.

These claims paint a picture of quantum space-time that is much simpler than should have been expected, including for instance a smooth and semiclassical transition through high curvature: "Yet, in all cases where the detailed evolution of *quantum* states has been carried out, effective equations have provided excellent approximations to the full quantum evolution of LQC [loop quantum cosmology] even in the deep Planck regime, provided the states are semi-classical initially in the low curvature regime" (emphasis in [1])[1] or "Indeed, the effective equations even provide an analytical expression of the maximum density $\rho_{\max}$ whose value is in complete agreement with the exact numerical simulations!" [1]. Later in the same review, regarding models with positive spatial curvature, "For a universe that undergoes a classical recollapse at $\sim 1\mathrm{Mpc}$, a state that nearly saturates the uncertainty bound *initially*, with uncertainties in $\hat{p}_{(\phi)}$ and $\hat{V}|_\phi$ spread equally, the relative dispersion in $\hat{V}|_\phi$ is still $\sim 10^{-6}$ after some $10^{50}$ cycles" (emphasis in [1]). In the case of a negative cosmological constant, "Because the level spacing between the eigenvalues of $\Theta'_\Lambda$ is not exactly periodic, there is a slight spread in the wave function from one epoch to the next. However, this dispersion is *extremely* small. For a macroscopic universe with $\Lambda = 10^{-120}m_{\mathrm{Pl}}^2$, the initially minute dispersion doubles only after $10^{70}$ cycles!" (emphasis in [1], where $\Lambda = 10^{-120}m_{\mathrm{Pl}}^2$ should be $|\Lambda| = 10^{-120}m_{\mathrm{Pl}}^2$).

---

[1] All quotations from [1] given in this paper refer to the second preprint version.

For a positive cosmological constant, the evolution operator with respect to a scalar-field internal time, referred to in [1] as $\Theta_\Lambda$, is not essentially self-adjoint, unlike the generator $\Theta'_\Lambda$ for a negative cosmological constant. The statement "It is, however, quite surprising that the evolution of such semi-classical states is largely independent of the self-adjoint extension chosen" [1] again suggests that quantum effects are severely suppressed for reasons unknown to [1]: "But the precise reason behind the numerically observed robustness of the quantum evolution is far from being clear and further exploration may well lead one [sic] an interesting set of results on sufficient conditions under which inequivalent self-adjoint extensions yield nearly equivalent evolution of semi-classical states."

Or, regarding inflationary potentials, a procedure is first set up by "Now, we are interested in states which are sharply peaked on a general relativity trajectory in the regime in which general relativity is an excellent approximation to LQC. The question is: if we evolve them backward in time using [a holonomy-modified wave equation] (4.22), do they remain sharply peaked on the corresponding solution of the effective equations across the bounce? To answer this question, we need a sufficiently long 'time' interval so that the state can evolve from a density of, say, $\rho = 10^{-4}\rho_{\max}$ where general relativity is a good approximation, to the putative bounce point and then beyond." It is then claimed to imply that "wave functions continue to remain sharply peaked on effective trajectories at and beyond the bounce, independently of the choice of the self-adjoint extension."

These statements, all quoted from a single review which, upon close reading, reveals several troubling features, have often been made within a specific school of thought within loop quantum cosmology, but they have never been explained based on general physics principles. While the review [1] is by now rather old, it has been foundational and is still widely believed to be valid in the very recent literature. Most of the latter did not question the claims made in [1] but rather built on them. A detailed analysis of [1] therefore remains timely. Other critical viewpoints have occasionally been presented, such as [2,3] which discuss observational questions and the robustness of bounces but do so within the setting described in [1]. Others, such as [4,5], focus on technical questions of a specific Hilbert-space representation. Our discussion here will be broader (and therefore much more damning), pointing out several violations of general physics principles such as the ubiquity of quantization ambiguities, the domain of effective field theory, and the condition of general covariance.

In terms of specific constructions, the name "loop quantum cosmology" in current parlance does not refer to a single theory but rather describes a collection of different frameworks built on the same basic motivation, originally given in [6,7]. The most optimistic, and at the same time puzzling, claims have been made in what will be referred to here as the "Ashtekar school" because it was initiated by a series of papers starting with [8]. These papers and various follow-up studies, summarized in [1], introduced several simplifying assumptions which initially were not deemed highly restrictive or misleading. One example is the overly semi-classical nature of evolved states, as already quoted, which is presented as a surprising result of the general framework but, in fact, represents only one of several consequences of an erroneous assumption in the very foundations of the school.

As another example, consider several attempted explanations of singularity resolution. The abstract of [1] claims that "In particular, quantum geometry creates a brand new repulsive force which is totally negligible at low space-time curvature but rises rapidly in the Planck regime, overwhelming the classical gravitational attraction" and later "The key difference between the WDW [Wheeler–DeWitt] theory and LQC is that, thanks to the quantum geometry inherited from LQG [loop quantum gravity], LQC has a novel, built-in repulsive force." The review continues by saying that "For matter satisfying the usual energy conditions any time a curvature invariant grows to the Planck scale, quantum geometry effects dilute it, thereby resolving singularities of general relativity" or, more explicitly, "Thus, the LQC resolution of the big-bang singularity can evade the original singularity theorems of general relativity even when matter satisfies *all* energy conditions because Einstein's equations are modified due to quantum gravity effects" (emphasis in [1]). These claims are puzzling because the broader singularity theorems of general relativity do not require specific dynamics but only use properties of Riemannian geometry such as the geodesic deviation equation. Singularity

removal, therefore, cannot be explained simply by a new force, which would be a dynamical feature. Or, if there is a new force that is able to modify the geodesic deviation equation, it would also have to render space-time geometry non-Riemannian. No such possibility has been considered by the Ashtekar school.

A deeper analysis reveals that these and other claims made by the Ashtekar school, percolating through the entire manifesto [1], are not only incorrect but also hint at much deeper problems of the framework. Valuable lessons can be learned from such an analysis, not only for loop quantum cosmology itself but also for quantum cosmology in broader terms, as well as for the full theory of loop quantum gravity or other discrete approaches to quantum gravity. They also apply to various black-hole models that are based on replacing the singularity with a bouncing interior within the horizon. Some observations put together here for a general critique have already been made before, for instance the importance of infrared renormalization [9], the dependence of bounce robustness on a choice of dynamical representation [10], or the lack of covariance [11]. In addition to briefly reviewing these observations, we provide here a comprehensive case that reveals the extensive scope of problems in loop quantum cosmology, in particular in the Ashtekar school, and we present an outline of steps that would be required to address these issues.

We briefly summarize the main problems to be discussed: (i) the Ashtekar school bases its constructions on the erroneous assumption that large comoving regions may be assumed to be homogeneous in the early universe. As a justification, an appeal is made to the Belinskii–Khalatnikov–Lifshitz (BKL) scenario [12] without noting that this scenario is valid only asymptotically close to a singularity and does not provide a generic lower bound for the size of homogeneous regions. As we will show in Section 2, this mistake implies that quantum effects and their associated quantization ambiguities are severely underestimated. (ii) The Ashtekar school implicitly makes strong assumptions about the availability of effective descriptions, thereby rendering any results based on them non-generic. In particular, the Ashtekar school assumes that a single effective theory can be used throughout a wide range of vastly different scales, including the late and the very early universe (and even through a bounce). In addition, the school assumes that a certain parameter calculated in a fundamental theory (the so-called "area gap") can directly be inserted in the equations of an effective theory even though no derivation of the latter from the former theory exists at present. These problems will be discussed in Section 3. (iii) As shown in Section 4, the Ashtekar school has made several incorrect claims about the realization of general covariance in its models.

## 2. Quanta of Loop Quantum Cosmology

Key assumptions made by the Ashtekar school downplay the importance of quantum corrections and quantization ambiguities in loop quantum cosmology. The former is related to an incorrect interpretation of the averaging volume used to define homogeneous minisuperspace models; the latter, to an implicit choice of a dynamical representation which, as it turns out, helps to make bouncing solutions more prevalent. Quantization ambiguities are also downplayed by an incorrect understanding of effective theory, which will be the main topic of the next section. (We note that ambiguities have occasionally been discussed in the context of the Ashtekar school, for instance the recent revival [13] of effective theories that include higher-order contributions from the so-called Lorentzian part of the Hamiltonian constraint [7,14–16]. However, such a phenomenological analysis takes for granted that there is a semiclassical bounce and ignores the possibility that quantization ambiguities may challenge the very existence of a bounce as well as the validity of certain effective theories.)

### 2.1. Loop Quantum Classicality

Loop quantum cosmology in all its forms uses connections and triads as basic variables, given in isotropic and homogeneous models by

$$A_a^i = c\delta_a^i \quad , \quad E_i^a = p\delta_i^a \tag{1}$$

with time-dependent functions $c(t)$ and $p(t)$. Inserting these fields in the symplectic term of the full gravitational action,

$$\frac{1}{8\pi G} \int_{\mathcal{V}} \dot{A}_a^i E_i^a \mathrm{d}^3 x = \frac{3V_0}{8\pi G} \dot{c} p, \tag{2}$$

integrated over a finite spatial region $\mathcal{V}$ with coordinate volume $\int_{\mathcal{V}} \mathrm{d}^3 x = V_0$, implies the Poisson bracket

$$\{c, p\} = \frac{8\pi G}{3V_0} . \tag{3}$$

(Without loss of generality, we set the Barbero–Immirzi parameter [17,18] to the value $\gamma = 1$.) This bracket depends on the arbitrary coordinate volume $V_0$ of the averaging region $\mathcal{V}$. Note that $V_0$ depends on two conceptually different properties, the choice of the region $\mathcal{V}$ as well as spatial coordinates $x$. Assuming that spatial coordinates have been fixed at an initial stage, we will be concerned only with the former property, for instance when we restrict $\mathcal{V}$ to subregions in a process of infrared renormalization that implies shrinking $V_0$ in a collapsing universe.

For simplicity, facilitating a direct comparison with [1], we will transform these basic variables to the Hubble parameter $\mathcal{H} = c/\sqrt{|p|}$ and the oriented volume $v = |p|^{3/2}\mathrm{sgn}p$:

$$\{\mathcal{H}, v\} = \frac{4\pi G}{V_0} . \tag{4}$$

The next step introduces modifications motivated by a discrete structure of space in loop quantum gravity. While loop quantum gravity is a quantum theory of continuum fields, it leads to discrete spectra of geometrical operators such as the spatial volume [19–22]. The corresponding quantum representation [23,24] implies that the Hamiltonian constraint cannot be quantized directly in its classical form and has to be modified [14,25]. As we will indicate briefly in Section 4, the brackets of constraints then carry information about a modified space–time structure, indirectly related to the discreteness of geometrical spectra. The modifications to be discussed now, initially motivated by formal properties of a quantum representation, are therefore indicative of potential dynamical effects implied by discrete space.

In the isotropic context, instead of representing $\mathcal{H}$ and $v$ in the standard way of quantum mechanics, only finite shifts in $v$, generated by $\mathcal{H}$, are represented through holonomy operators

$$\hat{h}_\mu = \widehat{\exp(i\mu\mathcal{H})}, \tag{5}$$

where $\mu$ is a real parameter with units of length. It is often assumed that $\mu$ is close to the Planck length, based on common dimensional arguments, but a precise value remains to be derived from the full theory for instance through some suitable procedure of coarse-graining. The parameter $\mu$ is not a regulator (but a modifier) because it is not removed in the quantization procedure used in loop quantum cosmology. As we will discuss in Section 3, in a proper effective theory $\mu$ may depend on time through the volume.

The basic commutator then takes the form

$$[\hat{h}_\mu, \hat{v}] = -\frac{4\pi G\hbar\mu}{V_0} \hat{h}_\mu \tag{6}$$

and implies a $V_0$-dependent uncertainty relation

$$\Delta_\mu \mathcal{H} \Delta v \geq \frac{2\pi G\hbar}{V_0} . \tag{7}$$

In the absence of an operator directly for $\hat{\mathcal{H}}$, we define $\mathcal{H}$-fluctuations as

$$\Delta_\mu \mathcal{H} = \frac{\Delta \widehat{\sin(\mu\mathcal{H})}}{\mu \langle \widehat{\cos(\mu\mathcal{H})} \rangle} . \tag{8}$$

In the late universe, large-scale homogeneity implies that the averaging volume, $V = V_0|v|$, is large. Dividing (7) by $\langle |\hat{v}| \rangle$, we therefore have

$$\Delta_\mu \mathcal{H} \frac{\Delta V}{\langle \hat{V} \rangle} \geq 2\pi \frac{\ell_P^2}{\langle \hat{V} \rangle} \tag{9}$$

with a tiny minimum

$$\left( \left( \langle \hat{V} \rangle^{1/3} \Delta_\mu \mathcal{H} \right) \frac{\Delta V}{\langle \hat{V} \rangle} \right)_{min} = 2\pi \frac{\ell_P^2}{\langle \hat{V} \rangle^{2/3}} \ll 1 \tag{10}$$

for the product of dimensionless fluctuations, $\langle \hat{V} \rangle^{1/3} \Delta_\mu \mathcal{H}$ and $\Delta V / \langle \hat{V} \rangle$. It is, therefore, possible to find late-time states which are very semiclassical, with tiny relative fluctuations of all basic operators. Such states easily stay sharply peaked for a long time, which is not surprising because they represent macroscopic objects with huge volumes $\langle \hat{V} \rangle$.

However, the same arguments cannot be used at early times, close to the Planck regime. If we follow the collapse of an initially large-scale homogeneous universe, while $V$ shrinks, the structure forms even within a co-moving volume $\mathcal{V}$ of constant coordinate size $V_0$. Once inhomogeneity within $\mathcal{V}$ is appreciable, a smaller region should be selected if the collapse process is still to be tracked using a homogeneous model. The scale of homogeneity is therefore progressively reduced as time goes on, without any lower bound on possible $V_0$ in the classical theory. Nevertheless, there is still a role played by homogeneous dynamics even close to a spacelike singularity because the BKL scenario suggests that the generic approach to a spacelike singularity is locally homogeneous: close to a spacelike singularity, time derivatives in Einstein's equation dominate over spatial derivatives, implying a dynamics approximated by homogeneous models. Nevertheless, the scale of homogeneity is small, without any general lower bound. Only the dynamics but not the total homogeneous space of a The Bianchi model can be used as an approximation. In particular, a small averaging region $\mathcal{V}$, or a small $V_0$ if spatial coordinates have been fixed, should be used in order to adjust the dynamics to be of BKL type rather than the large-scale homogeneous late-time form. Because the BKL scenario does not set a lower bound on the scale of homogeneity, it does not justify any assumption that $V$ should remain much greater than the Planck volume; it might even be possible to have $V \ll \ell_P^3$ well before Planckian curvature has been reached. (The BKL scenario, being purely classical, assigns no major role to the Planck length.) The minimal product of quantum fluctuations implied by (9) is then no longer small, and states are very quantum.

The Ashtekar school, however, works with the same, large $V_0$ throughout the entire evolution, thereby suppressing quantum effects in the Planck regime. To be sure, the averaging volume $V$, given by a constant $V_0$ times the scale factor cubed, does shrink in a collapsing model. However, because inhomogeneity generically builds up in any collapsing co-moving volume, the region $\mathcal{V}$, and thus $V_0$, should be shrunk at the same time in order to maintain the approximation by a homogeneous model. This reduction of the scale of homogeneity does not follow from solutions of minisuperspace equations of motion, but rather from the implementation of infared renormalization in an effective theory that takes into account limitations of strict minisuperspace truncations.

There is a subtle mistake in statements such as "Indeed, according to the BKL conjecture, the behavior of the gravitational field as one approaches generic space-like singularities can be largely understood using homogeneous but anisotropic models. This makes the question of singularity resolution in those models conceptually important" [1]. It is not the full space of a homogeneous model that is relevant here but only a tiny region in this space. Classically, there is no significant difference between these two interpretations, but the distinction is crucial in quantum cosmology because uncertainty relations such as (9) depend on the volume. Statements such as "For example, for a [closed] universe which grows to a maximum volume of $1Gpc^3$, the volume at the bounce is approximately $10^{117} \ell_{\rm Pl}^3$! That the bounce occurs at such a large volume may seem surprising at first. But what matters is curvature and density and these are *always* of Planck scale at the bounce" (emphasis in [1]) therefore misplace the wonderment. What is surprising is not the possibility of having a Planckian density at large volume, but rather the unmentioned assumption that a generic universe at Planckian density may still be described by a completely homogeneous space as large as $10^{117} \ell_{\rm Pl}^3$.

Sometimes, the Ashtekar school raises its own puzzling questions in this context, such as [1] "How do quantum gravity corrections manage to be dominant near the singularity in spite of the fact that the classical action is large? As we will see, the origin of this phenomenon lies in quantum geometry" and "For, in the path integral formulation quantum effects usually become important when the action is small, comparable to the Planck's constant $\hbar$, while the Einstein–Hilbert action along classical trajectories that originate in the big-bang is generically very large." In fact, it is wrong to assume that the action of a homogeneous model is large near the singularity because one should work with microscopic, not macroscopic $\mathcal{V}$. The quantum-geometry origin alluded to in the quote is simply a modification of the classical dynamics, which is unrelated to whether the action is large or small.

Although parameter estimates given in [1] rarely refer to the value of $V_0$ (which is coordinate dependent), they imply large values by the numbers assigned to the $V_0$-dependent momentum $p_\phi = V\dot\phi/N$ of a free massless scalar $\phi$ used for deparameterization: "The value of the supremum $\langle \hat\rho_\phi \rangle$ is directly determined by the area gap and is in excellent agreement with the earlier studies based on numerical evolution of semi-classical states. As an example, for semi-classical states peaked at late times in a macroscopic universe with $\langle \hat{p}_{(\phi)} \rangle = 5000\hbar$, the density at the bound [sic] already agrees with $\rho_{\rm sup}$ to 1 part in $10^4$. In the k=1 case, for the universe to reach large macroscopic sizes, $\langle \hat{p}_{(\phi)} \rangle$, has to be far larger. If we use those values here, then the density at the bounce and $\rho_{\rm sup}$ would be indistinguishable." Again, in the context of a negative cosmological constant $\Lambda$, "Let us consider Schrödinger states at a late time which are semi-classical, peaked at a point on a dynamical trajectory with a macroscopic value of $p_{(\phi)}$" such that "For macroscopic $p_{(\phi)}$, the agreement between general relativity and LQC is excellent when $\rho_{\rm tot} = \rho + \rho_\Lambda \ll \rho_{\rm max}$: departures are significant only in Planck regimes. In particular, the wave packet faithfully follows the classical trajectory near the recollapse. Thus, again, LQC successfully meets the ultraviolet and infrared challenges discussed in section I." These statements show that the same, macroscopic averaging region is used throughout the entire evolution from low curvature (where the universe is indeed large-scale homogeneous to a good approximation) all the way to Planckian curvatures where the only justification for using homogeneous dynamics, given by the BKL scenario, requires microscopic homogeneous regions.

The erroneous application of macroscopic averaging volumes even at high curvature is then used to claim robustness of loop quantum cosmology with respect to quantization ambiguities: "These examples illustrate that, even though a priori it may seem that there is considerable freedom in defining the Hamiltonian constraint, one can introduce well motivated criteria that can serve as Occam's razor. The two LQC examples we discussed bring out four important points: (i) internal coherence, a good ultraviolet behavior and the requirement that quantum dynamics should not lead to large deviations from general relativity in tame regimes, already constitute powerful constraints; [...] (iv) Although the totality of requirements may seem oppressively large, they *can be* met if one follows a well motivated path that is conceptually well-grounded" (emphasis in [1]; note that the reference to Occam's razor

in the context of quantization ambiguities is misplaced because its proper use refers to the number of assumptions invoked for a specific explanation, while quantization ambiguities do not constitute independent assumptions). A closely related problem will be discussed in more detail in the next section: the claim that a single effective theory should be used through a wide range of curvature or energy scales.

The issue of cosmic forgetfulness [26,27] (see also [28] in the context of group-field cosmology [29–32]) is also misinterpreted by the Ashtekar school as a consequence of several misconceptions, in particular about the role of the averaging region. The authors of [1] quote a result from [33]:

$$\text{"}|\sigma_+ - \sigma_-| \le 2\sigma_*, \tag{11}$$

where

$$\sigma_\pm = \langle \Delta \ln \frac{\hat{V}|_\phi}{2\pi\gamma\lambda\ell_{\text{Pl}}^2} \rangle_\pm, \qquad \sigma_* = \langle \Delta \ln(\frac{\hat{p}_{(\phi)}}{\sqrt{G\hbar}}) \rangle\text{"}, \tag{12}$$

and $\lambda^2 = 4\sqrt{3}\pi\gamma\ell_{\text{Pl}}^2$. The inequality is supposed to restrict the difference of relative volume fluctuations, $\sigma_\pm$, at times well before $(-)$ and well after $(+)$ the bounce. As usual, the authors of [1] "begin with a semi-classical state in the distant past well before the bounce" where a large averaging region can be used. Then, "To make the discussion more concrete let us suppose that, when the hypothetical universe under consideration has a radius equal to the observable radius of our own universe at the CMB time, it has the same density as our universe then had. For such a universe $\langle \hat{p}_{(\phi)} \rangle \approx 10^{126}$ in Planck units. Thus, the coefficient on the right side of (3.30) [our (11)] is $\sim 10^{-124}$!" However, $p_\phi$ depends on $V_0$, and therefore gets successively smaller through infrared renormalization as the bounce is approached (notwithstanding the fact that $p_\phi$ is a constant of motion in a free-scalar model based on a fixed $V_0$). The estimate given in [1] therefore is not applicable in a discussion of relationships between pre and post bounce values of volume fluctuations. (The right-hand side of (11) is determined by quantum fluctuations, which generically depend on $V_0$; see (7).) For a further discussion of cosmic forgetfulness, see [34].

In fact, an argument is often used to suggest that a limit $V_0 \to \infty$ must be taken, removing any non-zero lower bound on quantum fluctuations in (7). This limit has been suggested because a classical isotropic model without spatial curvature can be described by an infinite space $\mathbb{R}^3$, in which any finite averaging region $\mathcal{V}$ is only one small part. The restriction of space to a finite region is then considered an infrared regulator, which, as is argued, should be removed after quantization by taking the limit $V_0 \to \infty$. According to [1], "As we saw in section II, a natural strategy is to introduce an infrared regulator, i.e., a cell $\mathcal{C}$, and restrict all integrations to it. But we have the rescaling freedom $\mathcal{C} \to \beta^3 \mathcal{C}$ where $\beta$ is a positive real number. How do various structures react to this change? At the classical level, we found in section II A that, although the symplectic structure and the Hamiltonian transform non-trivially, physics—the equations of motion for geometry and matter fields—are all invariant under the rescaling. What about the quantum theory?" A regulator implemented in the classical theory does not leave physics, that is, the classical solution space, invariant; this is why a regulator ultimately has to be removed. Sometimes, a regulator leaves equations of motion invariant, as indicated in the quote, but then restricts the solution space by imposing boundary conditions, as when a field is put in a box as a proper infrared regulator. However, choosing a finite $V_0$ in a homogeneous model based on an infinite space $\mathbb{R}^3$ does not impose any boundary conditions on homogeneous solutions. Changing the value of $V_0$ does not in any way affect the solution space of a classical homogeneous model. (The Hamiltonian should of course change under this transformation because its matter part determines the energy contained in a region of size $V_0$, which does depend on $V_0$.) Varying $V_0$ is, therefore, a classical symmetry, not the choice of a regulator. Such a symmetry may or may not be broken by quantum effects.

Turning first to the so-called $\mu_o$-scheme, [1] continues by saying "However, a closer examination shows that this dynamics has several inadmissible features. First, the energy density at which bounce

occurs scales with the change $\mathcal{C} \to \beta^3 \mathcal{C}$ in the size of the fiducial cell. […] But the density at the bounce should have a direct physical meaning that should not depend on the size of the cell. Moreover, if we were to remove the infrared regulator in this final result, i.e., take the limit in which $\mathcal{C}$ occupies the full $\mathbb{R}^3$, we would find $\rho_{\max}^{(\mu_o)} \to 0$!" Mistaking scaling independence after quantization as a "theoretical coherence criterion", [1] observes that it can be bypassed "by restricting oneself to the spatially compact $\mathbb{T}^3$ topology where one does not need a fiducial cell at all." Here, [1] erroneously suggests that homogeneity of the entire space should be used in a study of near-singular dynamics. The final conclusion, "Suppose that, when its radius equals the observable radius of our own universe at the CMB time, it has the same density as our universe then had. For such a universe $p_{(\phi)} \approx 10^{126}$ in Planck units so the density at the bounce would be $\rho_{\max}^{(\mu_o)} \approx 10^{-32}$gm/cm$^3$!" [1], then makes the mistake of assuming a full homogeneous space as well as a constant, macroscopic $\mathcal{V}$ throughout a wide energy or curvature range. An extrapolation from CMB scales to the Planck scale within a single homogeneous effective description is illegitimate.

These arguments are flawed for several reasons. In particular, restricting homogeneous space, be it finite or infinite, to a region $\mathcal{V}$ does not constitute regularization because it modifies neither equations of motion nor the solution space of the theory. Classically, we can work with any finite $V_0$ and obtain the same predictions for observables. Changing $V_0$ is, therefore, a symmetry of the classical theory unrelated to setting a regulator that would have to be removed. This symmetry is broken by quantum effects when fluctuations are included, as shown by (7) which is not covariant with respect to changing $V_0$. If it is supposedly possible to counter this broken symmetry by choosing a specific value of $V_0$, such as the volume of a torus, it must be justified by additional arguments that have not been provided by the Ashtekar school. Because the symmetry is broken at the quantum level, any such argument should refer to specific quantum physics.

In fact, $V_0$ does not set an infrared regulator but rather a running infrared scale that determines the range in which a homogeneous (or perturbatively inhomogeneous) description can be used. It is then not surprising that quantum effects computed in a homogeneous model depend on the scale ($V_0$) on which homogeneity is realized. This running scale changes from large (truly infrared) values at late times to small (actually ultraviolet) values at early times, described within a BKL scenario. Connecting infrared physics to ultraviolet physics in this manner requires much care, which is completely neglected if one works with a fixed, large $V_0$ throughout all evolution. How to set up a valid effective field theory of loop quantum cosmology will be discussed in more detail in the next section. For now, we will consider the small-$V_0$ regime in more detail in order to see implications for the possibility of non-bouncing solutions.

### 2.2. Loop Quantum Serendipity

Proceeding with how a specific version of loop quantum cosmology is set up, we return to the basic commutator (6) and see how it may affect the dynamics. To this end, we should find a dynamical representation in which not only $\hat{h}_\mu$ and $\hat{V} = V_0|\hat{v}|$ operate on wave functions, respecting (6), but also a suitable Hamiltonian that encodes the dynamics of isotropic cosmological models. It turns out that the principles of loop quantum cosmology can accommodate a large variety of inequivalent representations, some but not all of which have a strictly positive volume on dynamical solutions in a free scalar model. Such a representation has serendipitously been chosen by the Ashtekar school, thereby overstating the prevalence of bouncing solutions.

### 2.2.1. Loop Quantum Cosmology as a Discrete Affine Theory

Dynamics requires time, which exposes loop quantum cosmology to the problem of time in quantum gravity [35–37]. The Ashtekar school does not solve this problem but rather evades it by working with deparameterization in a fixed choice of internal time, given by a free, massless scalar $\phi$ with momentum $p_\phi$. (Briefly, deparameterization does not lead to time reparameterization invariance after quantization because different choices of internal time that may exist in a single model generically

imply inequivalent quantum corrections [38,39].) The Friedmann equation for spatially flat models then gives rise to the Hamiltonian constraint

$$C = -\frac{3}{8\pi G} V \mathcal{H}^2 + \frac{1}{2}\frac{p_\phi^2}{V} = 0 \,. \tag{13}$$

Applying deparameterization at the classical or quantum level, we obtain $\phi$-evolution generated by a Hamiltonian operator

$$\hat{H}_\phi = -\hat{p}_\phi = \pm\sqrt{\frac{3}{4\pi G}}|\widehat{V\mathcal{H}}| \,. \tag{14}$$

A straightforward quantization of this Hamiltonian would be given in terms of the dilation generator $\hat{D} = \widehat{V\mathcal{H}}$, which is well-defined and self-adjoint in a quantization of the positive real line, $V > 0$. It is therefore a basic operator in affine quantum cosmology [40–44], in which the volume is restricted to positive values. In loop quantum cosmology, both signs are allowed for the oriented volume $v$, taking into account the orientation of space. Nevertheless, the dilation generator plays a role because it happens to be the Hamiltonian (14) in a simple deparameterized model.

However, loop quantum cosmology does not provide an operator $\hat{\mathcal{H}}$. A more involved quantization of (14) is therefore required, making use of holonomy operators (5) with some non-zero $\mu$ (a quantization ambiguity). With this modification, the loop-quantum-cosmology version of (14) can be interpreted as a discrete affine quantization of isotropic cosmology. Instead of basic operators $\hat{V}$ and $\hat{D}$ as in affine quantum cosmology, it uses basic operators $\hat{v}$ together with

$$\hat{J}_\mu = \mu^{-1}\hat{v}\hat{h}_\mu \quad , \quad \hat{J}_\mu^\dagger = \mu^{-1}\hat{h}_{-\mu}\hat{v} \,, \tag{15}$$

where the factor of $\hat{v}$ demonstrates the kinship to affine quantum cosmology, while the use of holonomy operators, quantizing periodic functions of $\mathcal{H}$, provides discreteness of $v$. (In what follows, we will suppress the subscript $\mu$ on $\hat{J}$ in order to avoid overloading the notation.) Using self-adjoint linear combinations

$$\hat{J}_+ = \mathrm{Re}\hat{J} = \frac{1}{2}\left(\hat{J} + \hat{J}^\dagger\right) \quad , \quad \hat{J}_- = \mathrm{Im}\hat{J} = \frac{1}{2i}\left(\hat{J} - \hat{J}^\dagger\right) \tag{16}$$

the basic operators $(\hat{v}, \hat{J}_+, \hat{J}_-)$ can be seen to be generators of the Lie algebra $\mathrm{sl}(2, \mathbb{R})$ with relations [45]

$$[\hat{v}, \hat{J}_+] = -\frac{4\pi i \ell_{\mathrm{P}}^2 \mu}{V_0}\hat{J}_- \quad , \quad [\hat{v}, \hat{J}_-] = \frac{4\pi i \ell_{\mathrm{P}}^2 \mu}{V_0}\hat{J}_+ \quad , \quad [\hat{J}_+, \hat{J}_-] = \frac{4\pi i \ell_{\mathrm{P}}^2}{\mu V_0}\left(\hat{v} - \frac{2\pi \ell_{\mathrm{P}}^2 \mu}{V_0}\right) \,. \tag{17}$$

(The shift of $\hat{v}$ in the last commutator can be eliminated by redefining the volume.) In simple models of loop quantum cosmology, this Lie algebra is dynamical because $|\hat{J}_-|$ can be used as a quantization of the Hamiltonian (14) in the deparameterized free-scalar model.

Loop quantum cosmology, therefore, replaces the Lie algebra of the affine group of $\mathbb{R}$ by $\mathrm{sl}(2, \mathbb{R})$. Unlike the Lie algebra of the affine group, $\mathrm{sl}(2, \mathbb{R})$ has several different types of representations, which generically do not preserve the sign of $v$ [46]. As pointed out in [10], the Ashtekar school has implicitly selected a specific representation from the positive discrete series in which no transformations change the sign of $v$ or even map to $v = 0$ eigenstates. Bouncing solutions are therefore guaranteed, but only by an implicit sleight of hand. Had one chosen a representation from the continuous series (in which, in spite of the name, the $\hat{v}$-spectrum is still discrete), dynamical operators connecting the two signs of $v$ would have been included. (Therefore, even if one accepts the modification of the classical constraint for non-zero $\mu$, bounces are not guaranteed. The modification implies an $\mathrm{sl}(2, \mathbb{R})$-structure but does not select a specific representation.)

### 2.2.2. Loop Quantum Bounceology

In addition to the choice of a discrete-series representation, the Ashtekar school has made several assumptions that, with hindsight, make it easier for solutions to bounce. For this reason, its discussions are about a pre-supposed bounce and do not constitute an unbiased approach to possible outcomes of quantum cosmology. Another example is the application of a large averaging volume even close to the big bang, where large-scale homogeneity is not generic. Working with a large averaging volume downplays the role of quantum corrections, as already discussed, because one is then dealing with a macroscopic object. Next to correction terms from quantum fluctuations, tunneling is another quantum effect that is suppressed for macroscopic objects (large $\mathcal{V}$) but can easily happen for microscopic objects (small $\mathcal{V}$). In particular, even if discreteness of the volume (or a "brand new repulsive force" [1]) may build up a barrier around $v = 0$, it is conceivable that a small homogeneous patch of a quantum universe can tunnel through it and encounter a singularity, in contrast to a large-scale homogeneous macroscopic patch.

Bounce statements are often based on upper bounds derived for the matter density on suitably generic solutions: "One can show that *all* quantum states (in the dense domain of the volume operator) undergo a quantum bounce in the sense that the expectation value of the volume operator has a non-zero lower bound in any state. More importantly, the matter density operator $\hat{\rho}|_\phi$ has a *universal upper bound* on $\mathcal{H}_{\mathrm{phy}}$ and, again, it coincides with $\rho_{\mathrm{max}}$ [from effective equations]." (emphasis in [1]). However, such bounds can be obtained even for non-bouncing quantum solutions on which the oriented volume $\hat{v}$ attains zero expectation value and becomes negative, while having non-zero variance. Non-zero lower bounds for $\langle|\hat{v}|\rangle$, as opposed to $\langle\hat{v}\rangle$, are easily possible in this situation because the absolute value is, by definition, never negative, and must then have a positive expectation value in a spread-out distribution. This fact is obscured in [1] (and elsewhere) because the authors repeatedly write equations such as (3.20) in [1],

$$\text{``}(F, \hat{v}F)_{\mathrm{phy}} = \frac{4\lambda}{\sqrt{12\pi G}} \int_{-\infty}^{\infty} \mathrm{d}x \left|\frac{\partial F(x_+, \phi)}{\partial x}\right|^2 \cosh(\sqrt{12\pi G}x)\text{''} \tag{18}$$

for the physical expectation value of the oriented volume, called $\hat{v}$ in [1], in a generic state given by the function $F(x_+, \phi)$ where $x_+ = \phi + x$, and $\lambda^2 = 4\sqrt{3}\pi\gamma\ell_{\mathrm{Pl}}^2$. The right-hand side is always positive, leading to the mentioned non-zero lower bound of the volume expectation value. However, it is not the expectation value of the oriented volume $\hat{v}$ written on the left-hand side, which would be more complicated in this setting; it may certainly be zero or negative. The volume expectation value on the right-hand side of (18) and the resulting lower bound of the volume are guaranteed to be non-zero simply because they are computed for a positive operator with non-zero fluctuations.

The origin of the lower bound therefore is not some mysterious repulsive force, as repeatedly claimed in [1], but quantum fluctuations which are present also in the Wheeler-DeWitt or any other quantization. The difference between the loop and Wheeler-DeWitt quantizations is only that the dynamics is modified in the former case, such that the minimal volume is reached at a finite value of $\phi$. In a Wheeler–DeWitt quantization, as in the classical model, the minimal volume is reached for $\phi \to \pm\infty$, a limit in which volume fluctuations can, and do, reach zero and no longer present an obstacle to a zero volume expectation value in the same limit. The attempted contrast between loop quantum cosmology and Wheeler-DeWitt quantum cosmology given in [1] is therefore misleading: "Thus, in the WDW theory, a state corresponding to a contracting universe encounters a big-crunch singularity in the future evolution, and the state corresponding to an expanding universe evolves to a big-bang singularity in the backward evolution", while in loop quantum cosmology "It is important to stress that the bounce occurs for arbitrary states in sLQC at a positive value of $\langle\hat{V}_{\phi_B}\rangle$ and the resolution of the classical singularity is generic." Even if $\langle\hat{V}\rangle = V_0\langle|\hat{v}|\rangle$ remains non-zero, $\langle\hat{v}\rangle$ may be zero, reaching the classical singularity.

Similarly, upper bounds on the energy density can be traced back to quantum fluctuations, even on solutions on which $\langle \hat{v} \rangle$ reaches zero. The latter is not prohibited in suitable representations of $sl(2, \mathbb{R})$, in particular from the continuous series. Fluctuations are subject to several conditions, including uncertainty relations as well as reality conditions that follow from the identity

$$\mu^2 \hat{J}\hat{J}^\dagger = \hat{v}^2 \tag{19}$$

according to the definition (15). Taking an expectation value and using the basic commutators, we obtain

$$\mu^2 \left( \langle \hat{J}_+ \rangle^2 + \langle \hat{J}_- \rangle^2 + (\Delta J_+)^2 + (\Delta J_-)^2 \right) = \left( \langle \hat{v} \rangle - \frac{2\pi \ell_{\rm P}^2 \mu}{V_0} \right)^2 - \frac{4\pi^2 \ell_{\rm P}^2 \mu^2}{V_0^2} + (\Delta v)^2 . \tag{20}$$

(The $\mu$-dependent constants on the right are subject to quantization ambiguities.) Identifying $J_-$ with the loop version of the deparameterized Hamiltonian (14), up to a constant factor, which equals $p_\phi$, we can write the reality condition as

$$\left( V_0 \langle \hat{v} \rangle - 2\pi \ell_{\rm P}^2 \mu \right)^2 - \mu^2 V_0^2 \langle \hat{J}_+ \rangle^2 = \tfrac{4\pi G}{3} \mu^2 \left( \langle \hat{p}_\phi \rangle^2 + (\Delta p_\phi)^2 \right) + \mu^2 V_0^2 (\Delta J_+)^2 - (\Delta V)^2 + 4\pi \ell_{\rm P}^2 \mu^2 . \tag{21}$$

For large $\langle \hat{p}_\phi \rangle$ (in a macroscopic universe), the first term on the right-hand side dominates. Since it is always positive, $V_0 \langle \hat{v} \rangle$ cannot reach zero, and not even the small but non-zero $2\pi \ell_{\rm P}^2 \mu$. Therefore, zero is avoided by the oriented volume, not just by the positive volume $\langle \hat{V} \rangle = V_0 \langle |\hat{v}| \rangle$.

However, during collapse, the averaging region and therefore the value of $\langle \hat{p}_\phi \rangle$ must be successively reduced in order to be consistent with the BKL scenario. There is no general lower bound on $\langle \hat{p}_\phi \rangle$ in this situation, such that the right-hand side of (21) may be zero or negative, eliminating any restrictions on possible values of $\langle \hat{v} \rangle$. Fluctuations on the right-hand side of (21) are restricted by uncertainty relations, but only from below such that the negative term $-(\Delta V)^2$ may dominate over the other, positive contributions.

As shown in more detail in [10], uncertainty relations can be used to derive the inequality

$$\langle \hat{V}^2 \rangle \geq \frac{2\pi G}{3} \mu^2 \langle \hat{p}_\phi^2 \rangle , \tag{22}$$

again using the relationship between $p_\phi$ and $J_-$ in loop models. In a simple estimate of the scalar energy density, we may replace the classical expression $\rho_\phi = p_\phi^2 / (2V^2)$ with $\langle \hat{p}_\phi^2 \rangle / (2\langle \hat{V}^2 \rangle)$. For this expression, the lower bound on $\langle \hat{V}^2 \rangle$ implies an upper bound

$$\frac{\langle \hat{p}_\phi^2 \rangle}{2\langle \hat{V}^2 \rangle} \leq \frac{3}{4\pi G \mu^2} . \tag{23}$$

For $\mu$ close to the Planck length, this upper bound is close to the Planck density. It holds for all solutions, even those on which $\langle \hat{v} \rangle = 0$ is reached. A Planckian upper bound on the energy density therefore does not imply that the oriented volume avoids the classical singularity.

The estimate of $\rho_\phi$ by $\langle \hat{p}_\phi^2 \rangle / (2\langle \hat{V}^2 \rangle)$ is subject to quantum fluctuations, and it depends on how one defines a density operator. The precise upper bound may therefore change. However, it is important to note that a Planckian upper bound on the density or a lower bound on the positive volume can be derived using only equations for quantum fluctuations, such as the reality condition and uncertainty relations. These conditions depend on the algebraic structure of the model, encoded in basic commutators, and are therefore different for Wheeler–DeWitt quantizations in which such bounds do not exist. There is certainly a difference between loop quantization and Wheeler–DeWitt quantization regarding the quantum nature of singularities, but it is implied by the behavior of quantum fluctuations rather than new repulsive forces in the former case. In particular, the Planckian value of the upper

density bound is only indirectly related to quantum geometry through the discreteness of $\hat{\vartheta}$ in a loop representation, which modifies the algebraic structure.

## 3. Effective Field Theory

Loop quantum cosmology cannot be a fundamental theory because the universe is not exactly homogeneous. It can, therefore, be claimed to be valid only as an effective theory, to within some approximation. While the connection between loop quantum cosmology and the purportedly fundamental theory of loop quantum gravity remains loose, the general principles of effective field theory should be taken into account when one sets up and interprets models of loop quantum cosmology. Unfortunately, claims made by the Ashtekar school completely ignore these principles. Instead, they focus on formal aspects such as the construction of physical Hilbert spaces ("To address these key physical questions, one needs a physical Hilbert space and a complete family of Dirac observables at least some of which diverge at the singularity in the classical theory" [1]) which are important in fundamental theories but play a less clear-cut role in effective theories. Working with a fixed physical Hilbert space prevents the Ashtekar school from implementing a scheme of infrared renormalization in which one would use macroscopic $V_0$ at low curvature and microscopic $V_0$ at high curvature in accordance with the BKL scenario, and it has other drawbacks to which we turn in this section.

In particular, a single effective theory cannot be assumed to be valid throughout a wide range of regimes, stretching from low curvature to high curvature and possibly back to low curvature. However, the Ashtekar school declared this misconception to be one of their founding principles; while initial statements such as "Can one construct a framework that cures the short-distance limitations of classical general relativity near singularities, while maintaining an agreement with it at large scales?" [1] are innocuous because they leave some freedom in how one interprets "construct a framework," they quickly evolve into specific claims which use the principle of a single effective theory to rule out possible quantizations: "In a nutshell, while the singularity was resolved in a well-defined sense, the theory predicted large deviations from general relativity in the low curvature regime. [. . . ] When this is corrected, the new, improved Hamiltonian constraint again *resolves the singularity and, at the same time, is free from all three drawbacks of the $\mu_o$ scheme.*" (emphasis in [1]). The conclusion that "Already in the spatially homogeneous situations, the transition from $\mu_o$ to $\bar{\mu}$ scheme taught us that great care is needed in the construction of the quantum Hamiltonian constraint to ensure that the resulting theory is satisfactory both in the ultraviolet *and* infrared" (emphasis in [1]) shows an intimate relationship between a strong claim to rule out ambiguities and the erroneous assumption that a single effective theory with fixed parameters should be valid throughout a wide range of scales.

We have already seen that the infrared scale, given by the averaging volume $V_0$ of minisuperspace models, should be adjusted to the varying homogeneity scale in a collapsing and possibly re-expanding universe. Along with this adjustment, parameters of the effective theory, in general, have to be changed as well, or renormalized.

### 3.1. Ineffective Theory

The Ashtekar school not only assumes that a single effective theory with fixed parameters specifying the discreteness scale remains valid through a wide range of energy scales, it also uses this erroneous belief as a condition to eliminate quantization ambiguities. It may, of course, be possible that such parameter choices can be made in certain situations, and that they fulfill other conditions set up within the minisuperspace setting. However, regime-independent choices are not generic in an effective theory, unless they have been strictly derived from a fundamental theory (which, at present, is not possible in loop quantum cosmology). Conclusions drawn from the assumption of using a single effective theory, related mainly to robustness claims in loop quantum cosmology, are thus invalid. The Ashtekar school thereby downplays the range of quantization ambiguities.

In addition, the Ashtekar school has made several further mistakes related to effective treatment. For instance, it misinterprets certain quantum corrections that arise from the discreteness scale. We have already seen holonomy modifications as one consequence of spatial discreteness in loop quantum gravity. It so happens that these modifications can (sometimes) be expressed solely in terms of the energy density of matter, related to the Planck density. In general, however, corrections in an effective theory may also refer to the discreteness scale of an underlying fundamental theory. Another example in loop quantum cosmology is given by inverse-triad corrections [47,48]. It is difficult to derive the specific behavior of these corrections while the fundamental behavior remains poorly understood, but this fact does not allow one to ignore them in an effective theory.

Using only the length scales available in a simple minisuperspace setting, the Planck length $\ell_{\mathrm{P}}$, together with the averaging volume $V_0$, the Ashtekar school assumes that any correction related to a length scale, given by spatial discreteness, must be determined by the dimensionless parameter $\ell_{\mathrm{P}}^3/V_0$. It then concludes that these corrections are either meaningless (because $V_0$ is not a physical parameter) or zero (because $V_0$ should be sent to infinity, removing an infrared regulator): "Numerical simulations show that, if we use states with values of $p_{(\phi)}$ that correspond to closed universes that can grow to macroscopic size, and are sharply peaked at a classical trajectory in the weak curvature region, the bounce occurs at a sufficiently large volume that these inverse scale factor corrections are completely negligible" [1] in spatially closed models, while in spatially flat models, "If one rescales the cell via $\mathcal{C} \to \beta^2 \mathcal{C}$, for the classical function we have $|\nu|^{-1/2} \to \beta^{-1}|\nu|^{-1/2}$ while the quantum operator has a complicated rescaling behavior. Consequently, the inverse volume corrections now acquire a cell dependence and therefore do not have a direct physical meaning [...] What happens when we remove the infrared-regulator by taking the cell to fill all of $\mathbb{R}^3$? Thus, the right side of (4.12) [inverse-volume corrections] goes to 1." The misinterpretation of the infrared scale as a regulator is made explicit by "when the topology is $\mathbb{R}^3$, while we can construct intermediate quantum theories tied to a fiducial cell and keep track of cell dependent, inverse volume corrections at these stages, when the infrared regulator is removed to obtain the final theory, these corrections are washed out for states that are semi-classical at late times" in [1]. Both interpretations ignore the correct role of $V_0$ as a running infrared scale. Corrections referring to the spatial discreteness scale should also be running, but methods to derive the relevant parameters do not yet exist in loop quantum gravity. This fact constitutes an incompleteness of loop quantum cosmology at present; it does not mean that inverse-triad corrections are meaningless or negligible.

Even the supposedly simple holonomy modifications, which in some cases can be formulated solely in terms of the energy density without reference to $V_0$, have been implemented incorrectly in effective theories used by the Ashtekar school. The usual claim is that holonomy modifications work by replacing the classical $\mathcal{H}$ in the Friedmann equation or Hamiltonian constraint with a periodic function such as $\sin(\mu \mathcal{H})/\mu$ which can be represented by holonomy operators. There are two relevant choices in this modification, given by the value of $\mu$ and by the specific function of $\dot{a}$ (or the canonical connection component $c$) and $a$ in which the replacement is periodic. (There is also a choice in the specific periodic function, which need not be a sine. This choice depends on details of the dynamics, see for instance [7,13,15,16], rather than general considerations of the effective theory.)

The Ashtekar school fixes the first choice by declaring that $\mu$, which is related to the discrete step size in $v$ generated by holonomies as shift operators, should be given by the smallest non-zero area eigenvalue in loop quantum gravity: "Quantum geometry of LQG tells us that at each intersection of any one of the edges with $S_{12}$, the spin network contributes a quantum of area $\Delta \ell_{\mathrm{Pl}}^2$ on this surface, where $\Delta = 4\sqrt{3}\pi\gamma$. For this LQG state to reproduce the LQC state $\Psi_o(\mu)$ under consideration, ..." and "The size of the loop, i.e., $\bar{\mu}$ was arrived [at] using a semi-heuristic correspondence between LQG and LQC states. This procedure parachutes the area gap from full LQG into LQC; in LQC proper there is no area gap" [1] postulate a very direct relationship between parameters and states in the fundamental and effective theories. Calling the semi-heuristic procedure by a fancy name (parachuting) is not sufficient to overcome the lack of any justification for using a fundamental parameter directly in an

effective theory. The same review cautions that "Similarly, it is likely that, in the final theory, the correct correspondence between full LQG and LQC will require us to use not the 'pure' area gap used here but a more sophisticated coarse-grained version thereof, and that will change the numerical coefficients in front of $\bar{\mu}$ and the numerical values of various physical quantities such as the maximum density we report in this review. So, specific numbers used in this review should not be taken too literally; they only serve to provide reasonable estimates, help fix parameters in numerical simulations, etc." However, no justification is given that the "parachuted" values may justifiably be considered reliable estimates of properly coarse-grained values. In particular, in the presence of large dimensionless numbers, such as the number of individual patches in a discrete state for a given region, it is not clear that coarse-grained values should be close to fundamental parameters even in terms of orders of magnitude.

The second choice, using $\mathcal{H} = c/\sqrt{|p|}$ as opposed to some other phase-space function as the argument of holonomies, has several a-posteriori justifications, mainly given by (i) the implicit assumption that a single effective theory should be valid throughout a wide range of energy scales and (ii) the desire to have corrections independent of $V_0$: "Note that the full set of effective equations has two key properties: i) they are free from infrared problems because they reduce to the corresponding equations of general relativity when $\rho \ll \rho_{\max}$; and (ii) they are all independent of the initial choice of the fiducial cell. Even though they may seem simple and obvious, these viability criteria are not met automatically but require due care in arriving at effective equations" [1]. The claim that "The functional dependence of $\bar{\mu}$ on $\mu$ on the other hand is robust" [1] is based on an erroneous interpretation of the averaging volume: "Consequently, the quantum Hamiltonian would have acquired a non-trivial cell dependence and even in the effective theory (discussed in section IV) physical predictions would have depended on the choice of $\mathcal{C}$." Again, "Reciprocally, if in place of $\bar{\mu} \sim \mu^{-1/2}$ we had used $\bar{\mu} = \mu_0$, a constant, the big-bang would again have been replaced by a quantum bounce but we would not have recovered general relativity in the infrared regime. Indeed, in that theory, there are perfectly good semi-classical states at late times which, even [sic] evolved backwards, exhibit a quantum bounce at density of water!"

We have already seen that both arguments in favor of using $\mathcal{H} = c/\sqrt{p}$ follows from an erroneous understanding of effective theory. The justification of the first choice—the value of $\mu$—is equally wrong, based on the claim that a parameter of a fundamental theory should directly determine an only conceptually related parameter of an effective theory. The argument of the modification function $\sin(\mu\mathcal{H})/\mu$ is therefore not fixed at all but, pending a derivation from loop quantum gravity, remains ambiguous in both the value of $\mu$ and the phase-space dependence (even if the sine function is accepted).

From its inception, the Ashtekar school has drawn a sharp dividing line between two quantization schemes, called the $\mu_0$ and $\bar{\mu}$-schemes. The $\bar{\mu}$-scheme corresponds to the modification just discussed, given by $\sin(\mu\mathcal{H})/\mu$ just writing $\bar{\mu}$ instead of $\mu$. The $\mu_0$-scheme has a similar ambiguity parameter $\mu = \mu_0$ but has periodicity in $c$ rather than $\mathcal{H} = c/\sqrt{|p|}$. However, in the absence of a strict derivation from loop quantum gravity, there is a large number of choices of phase-space functions, which moreover may differ in the various regimes in which effective theories may be valid. In keeping with common practice in cosmology, usually applied to the equation-of-state parameter $w = P/\rho$ of matter which need not be constant but may be assumed to be a piecewise-constant succession of different values for different power-laws of $\rho(a)$, one may parameterize holonomy modifications as [49]

$$h_{\mu,x}(p,c) = \frac{\sin(\mu|p|^x c)}{\mu|p|^x} \, . \tag{24}$$

The new ambiguity parameter $x$ formally plays the role of $w$. Different constants $x$ give different power laws for the $a$-dependence of the periodicity, but a single constant should generically be valid only in a certain regime of an effective theory, determined by a suitable density range. The two

distinct cases highlighted by the Ashtekar school, therefore, hide the appearance of a continuum of an ambiguity parameter characterizing different power-law behaviors.

The Ashtekar school claims that the $\mu_0$-scheme has been ruled out because it may imply bounces "at density of water" [1]. This claim, again, is based on the erroneous assumption that a single effective theory should be valid through a wide range of energy scales. Moreover, it assumes that $\mu_0$ is determined by the smallest non-zero area eigenvalue in loop quantum gravity, an unjustified assumption because it postulates that a parameter of a fundamental theory should directly provide the value of a parameter in an effective theory.

### 3.2. A Good Run of Loop Quantum Cosmology

How should an effective theory of loop quantum cosmology be set up? Deriving a strict effective theory from loop quantum gravity (or some other fundamentally discrete theory of quantum gravity) is challenging. However, several crucial ingredients are known at least from general considerations of effective theory applied to the context of quantum cosmology. In particular, the prevalent application of minisuperspace models in quantum cosmology can be meaningful, provided one takes into account the correct role played by the averaging region $\mathcal{V}$ as an infrared scale (but not an infrared regulator). This role can be inferred from analogous constructions [50] in the much better-understood case of scalar quantum field theory on a flat background space-time, where the Coleman–Weinberg potential [51] provides crucial insights.

In this setting, the infrared scale $\mathcal{V}$ determines the dividing line between an inner region, given by the physical scale of homogeneity, in which a minisuperspace model or such a model together with perturbative inhomogeneity may be used, and the surrounding space in which non-perturbative inhomogeneity (or a full fundamental theory) would have to be applied. The infrared scale is not constant but changes according to the physical scale of homogeneity: As inhomogeneity builds up in the inner region, $\mathcal{V}$ should be decreased, pushing more and more modes into the surrounding region in which a full field theory is applied. It is then not surprising, and certainly not inconsistent, for physical parameters derived in minisuperspace models, such as critical energy densities at which certain dynamical features happen, to depend on the infrared scale $\mathcal{V}$, or on the volume $V_0$ in a fixed set of spatial coordinates. This scale, after all, determines how many modes of the full theory have been approximated by a minisuperspace model. The collective quantum corrections implied by these modes (such as the infrared contribution to the Coleman–Weinberg potential in well-understood situations) then naturally depend on the infrared scale. Since the minisuperspace dynamics, or an early-universe model following the BKL scenario, reacts only to these quantum corrections but ignores corrections from strong inhomogeneity on larger scales, its dynamics depend on the infrared scale. Statements such as "But the density at the bounce should have a direct physical meaning that should not depend on the size of the cell" [1] are therefore inconsistent with a proper interpretation of the cell as a running infrared scale.

A detailed formulation requires a fully developed numerical (loop) quantum cosmology which not only evaluates the straightforward minisuperspace equations but at the same time tracks how inhomogeneity builds up through numerical relativity. The latter then tells one whenever the scale of $\mathcal{V}$ has to be adjusted in order to maintain a valid interior minisuperspace model. Parameters of this model run along with $\mathcal{V}$, in particular the values of $\mu$ and $x$ introduced in holonomy modifications (24) of loop quantum cosmology. How they run is a difficult question, which will eventually have to be answered by a derivation of effective equations from full loop quantum gravity, based on the form and dynamical behavior of discrete states. (Only limited information is currently available on this question in loop quantum gravity, for instance from studies of renormalization in this setting [52–57].) Even though the construction of such a complete effective theory remains out of reach, the sketch given here demonstrates that dependence of physical parameters on the infrared scale is only a problem of pure minisuperspace models in which the infrared scale is unknown. However, the scale would

be determined by the validity conditions for homogeneous approximations in small regions of an inhomogeneous simulation of quantum cosmology.

In early-universe cosmology, making use of the BKL scenario, the infrared scale will ultimately have to be moved into the ultraviolet, at which point the minisuperspace plus perturbative treatment is likely to lose its validity. Nevertheless, the approach to high curvature can be studied within this effective theory of quantum cosmology.

Attempts [58,59] have been made to derive parameters of effective equations, such as $\mu$ and $x$ in (24) (or $\bar{\mu}$) from spin-network states. However, current methods are insufficient for two reasons. First, they generally make use of regular lattice states, which implies an additional selection of states compared with full spin-networks. Secondly, they often work in a deparameterized setting, using the same free massless scalar $\phi$ applied in minisuperspace models. There is, therefore, an implicit choice of spatial slices, given by constant $\phi$, fixing the space-time gauge. Such a choice is valid only if the full theory of loop quantum gravity is slicing independent and covariant, which brings us to our last topic.

## 4. Covariance

The Ashtekar school often assumes that Riemannian space-time is realized in modified models, even if no attempt has been made to derive the correct space-time structure. (Recall also that this assumption is inconsistent with the claim that singularities are resolved even if positive-energy conditions are respected; see Section 1.) For instance, "one can show that this integral $[\int_0^\tau d\tau R_{ab}u^a u^b]$ is finite in the isotropic and homogeneous LQC, *irrespective of the choice of equations of state* including the ones which lead to a divergence in the Ricci scalar. Thus, the events where space-time curvature blows up in LQC are harmless weak singularities" (emphasis in [1]) refers to the Ricci tensor in a situation which is clearly modified compared with general relativity, such that the applicability of this classical object is unclear. The only argument in favor of classical space-time structures is erroneous, using Palatini-$f(R)$ models as supposed analog actions: "However, if one generalizes to theories where the metric and the connection are regarded as independent, one *can* construct a covariant effective action that reproduces the effective LQC dynamics" (emphasis in [1]). This claim is wrong: While it is possible to model some modifications in isotropic models of loop quantum cosmology using Palatini-$f(R)$ actions [60], any such theory applied to vacuum models, such as empty Bianchi models or the Schwarzschild interior described by a Kantowski–Sachs model, implies corrections to the Einstein–Hilbert action that merely amount to a modified cosmological constant [61]. It is, therefore, impossible for Palatini-$f(R)$ theories to describe the whole set of homogeneous models in loop quantum cosmology. Therefore, they cannot be used as a justification of covariance. (Notice that [61] was published several years before [1,60]. The key lesson from [61] in this context is implicitly contained already in the original paper [62] that introduced $f(R)$-gravity in 1970.)

Describing perturbative inhomogeneity, [1] mentions that some approaches use "suitable gauge fixing to make quantization tractable" which makes it impossible to obtain a full understanding of covariance, an off-shell property that is obscured by gauge-fixing. In the same context, [1] praises "The resulting Hamiltonian theories" because they "exhibit a clean separation between homogeneous and inhomogeneous modes" even though these modes play crucially inter-related roles in the hypersurface-deformation brackets [63,64] that govern covariance in the Hamiltonian formulation. (A background transformation of time and a small inhomogeneous normal deformation of a spatial slice commute only up to a spatial diffeomorphism on the slice, forming a semidirect product of Lie algebroids; see [65] for further discussions.) These modes should *not* be separated in a covariant model.

In [66], an attempt was made to extend the methods and ideas reviewed in [1] to black-hole models, using the well-known feature of the Schwarzschild or Kruskal space–time being spatially homogeneous inside the horizon. These constructions suffer from the same problem discussed here in cosmological models, assuming that a single effective theory with fixed parameters remains valid throughout a wide range of energy or curvature scales. (Further problems have been pointed out in [67,68].) In addition, the constructions presented in [66] have helped to reveal the full scope

of the covariance problem in models of loop quantum gravity. The model of [66] quantizes not only the homogeneous black-hole interior, but also applies homogeneous minisuperspace models to the exterior, using timelike slices which are homogeneous in a static space-time region. However, the same symmetry must be compatible with spherically symmetric, inhomogeneous, spacelike slices. For the vacuum models considered in [66], dilaton gravity theories present a powerful tool to analyze the possible dynamical equations consistent with holonomy-modified minisuperspace models on timelike slices.

In a more cosmological context, we may view the spherically symmetric exterior of a non-rotating black hole as a static version of Lemaitre–Tolman–Bondi (LTB) models

$$ds^2 = -M^2 dT^2 + S^2 dX^2 + R^2(d\vartheta^2 + \sin^2 \vartheta d\varphi^2), \tag{25}$$

with $T$-independent $M$, $S$ and $R$. As proposed in [66], one can turn the staticity condition, together with spherical symmetry, into a homogeneity condition on timelike slices, $X = $ const. Redefining the metric components according to

$$|p_c| = R^2 \quad , \quad p_b = MR \quad , \quad N = S, \tag{26}$$

and renaming the coordinates $T = x$ and $X = t$, we obtain a homogeneous model

$$ds^2 = -\frac{p_b^2}{|p_c|} dx^2 + N^2 dt^2 + |p_c|(d\vartheta^2 + \sin^2 \vartheta d\varphi^2) \tag{27}$$

of timelike slices, thus the unusual signs of $dx^2$ and $dt^2$. (The variables $p_b$ and $p_c$ correspond to densitized-triad components of the homogeneous model; see [69].) Equations of motion generated by a (modified) minisuperspace Hamiltonian constraint then determine how extrinsic-curvature components $b$ and $c$ are related to time derivatives of $p_b$ and $p_c$, as well as $N$.

Because a covariant, slicing-independent theory would have a single formulation with the same degrees of freedom for homogeneous timelike and inhomogeneous but static, spherically symmetric slices in the same region, the minisuperspace model shows what kind of degrees of freedom should be present. In particular, there should be fields only for a metric or a triad (but no scalar in the vacuum model), and they should be local because they are not accompanied by any additional degrees of freedom in a holonomy-modified minisuperspace model. General results from $1 + 1$-dimensional dilaton gravity [70] show that possible covariant and local modifications of spherically symmetric or LTB dynamics are much more restricted than modifications of homogeneous models, which in loop-quantum-cosmology descriptions of (27) have been applied rather liberally. In particular, covariance allows only the freedom of choosing a dilaton potential in the action or Hamiltonian—a function only of the coefficient $R$ but independent of $S$ and $M$ in (25).

As shown by an explicit transformation of equations of motion from modified homogeneous models to the coefficients defined in (25), however, the modifications implied by holonomies in minisuperspace models *cannot* be written in this way [11]. In particular, because the identification (26) relates $p_b$ and $N$ to $M$ and $S$, any holonomy modification, which by definition depends on $b$ or $c$ or both in a non-linear way, implies corrections in the spherically symmetric theory that depend on $M$ or $S$. This result is in conflict with the condition that the only available choice in a covariant $1 + 1$-dimensional dilaton model is given by the $R$-dependent dilaton potential. Therefore, the assumption that holonomy modifications are compatible with slicing independence is ruled out by contradiction.

The specific construction not only invalidates the black-hole models attempted in [66], it also highlights deep problems implied by holonomy modifications: even if one is interested only in cosmological models but not in a timelike homogeneous slicing as in [66], a covariant set of modifications should be implementable in a consistent way in all cases in which multiple slicings are possible, respecting the required symmetries. The observations of [11] then rule out holonomy

modifications as ingredients of slicing-independent space-time theories. They constitute a no-go theorem for the possibility of covariant holonomy modifications.

A possible solution to this problem is to implement holonomy modifications in an anomaly-free way which does not break any gauge transformations but may deform the classical structure of hypersurface deformations given in [63,64]. Consistent deformations are possible in spherically symmetric models with holonomy modifications [71–77], but they imply a non-classical space-time structure which is related to slicing independence only in some cases, and after field redefinitions [78,79]. The latter feature not only resolves the contradiction between holonomy modifications and covariance pointed out in [11], it also shows why singularities can be resolved in loop quantum cosmology even for matter obeying the usual energy conditions: not only the dynamics but also space-time structure become non-classical as a consequence of holonomy modifications, unhinging the mathematical foundation of singularity theorems. At the same time, deformations of space-time structure complicate any analysis of quantum space-time. In particular, they invalidate existing attempts to derive parameters of an effective theory of loop quantum cosmology from deparameterized models of full loop quantum cosmology, which implicitly assume that holonomy modifications are compatible with slicing independence; see also [80].

## 5. What's Left?

During the early stages of developments in a new research field, a certain laxness in rigor is often accepted in order to make some progress in spite of mounting obstacles. Preliminary observations then indicate how promising the new field may be, and where detailed investigations should be initiated in order to buttress the field's foundations. In this spirit, the Ashtekar school has led to useful results in quantum cosmology.

However, with the benefit of hindsight, it turned out that none of the specific assumptions made explicitly or implicitly by the Ashtekar school can be put on firm ground. Instead, as indicated by the account of the foundational developments in [1], the Ashtekar school took a wrong turn at a very early stage, and then found it impossible to correct its course. Crucial mistakes thereby were enshrined in its very foundations. These mistakes are not just inaccuracies in early models that might be amended by better approximations. Rather, they violate basic principles of a general physical nature and can be corrected only by eliminating the key ingredients that defined the Ashtekar school in the first place.

The large number of quotations from [1] given here has demonstrated the unfortunate prevalence of mistaken assumptions and claims. Let us finally collect the main erroneous statements made just in the Discussion section of [1]. As always in this critique, emphases in quoted passages are from [1].

- "As we saw in sections II–IV, although this brilliant vision [of Wheeler's] did not materialize in the WDW theory, it *is* realized in all the cosmological models that have been studied in detail in LQC. However the mechanism is much deeper than just the 'finite width of the wave packet': the key lies in the quantum effects of geometry that descend from full LQG to the cosmological settings. These effects produce an unforeseen repulsive force. Away from the Planck regime the force is completely negligible. But it rises *very* quickly as curvature approaches the Planck scale, overwhelms the enormous gravitational attraction and causes the quantum bounce." This statement overemphasizes the role of quantum geometry, while it ignores the fact that fluctuation effects explain much of the volume and density bounds obtained in loop quantum cosmology. Potential singularity resolution in loop quantum cosmology is therefore not dissimilar from what has been found in certain Wheeler–DeWitt-type quantizations; see for instance [81–84]. The mistake is repeated in "*In LQC the repulsive force has its origin in quantum geometry rather than quantum matter and it always overwhelms the classical gravitational attraction.*" which also overstates the prevalence of bounces, as non-bouncing solutions are possible in general loop quantum cosmology [10].

- "To obtain good behavior in both the ultraviolet *and* the infrared requires a great deal of care and sufficient control on rather subtle conceptual and mathematical issues" erroneously assumes that a single effective theory must be used through a wide energy or curvature range.

- In "Finally, it is pleasing to see that even in models that are *not* exactly soluble, states that are semi-classical at a late initial time continue to remain sharply peaked throughout the low curvature domain. [...] Initially this is surprising because of one's experience with the spread of wave functions in non-relativistic quantum mechanics." no surprise is warranted because in this regime one is dealing with a macroscopic object. Conversely, "The third notable feature is the powerful role of effective equations discussed in section V. As is not uncommon in physics, their domain of validity is much larger than one might have naively expected from the assumptions that go into their derivation. Specifically, in all models in which detailed simulations of *quantum* evolution have been carried out, wave functions which resemble coherent states at late times follow the dynamical trajectories given by effective equations even in the deep Planck regime" should have raised a severe warning. This statement hides the unmentioned (but wrong) assumption that macroscopic averaging regions may be used even "in the deep Planck regime."

- The claim that effective equations "arise from a (first order) covariant action" is incorrect because the proposed action fails in vacuum models. Similarly, more advanced recent versions which include a scalar field [85,86] fail to describe anisotropic or inhomogeneous modes in congruence with loop quantum cosmology [87–90].

- "The very considerable research in the BKL conjecture in general relativity suggests that, as generic space-like singularities are approached, 'terms containing time derivatives in the dynamical equations dominate over those containing spatial derivatives' and dynamics of fields at any fixed spatial point is better and better described by the homogeneous Bianchi models. Therefore, to handle the Planck regime to an adequate approximation, it may well suffice to treat just the homogeneous modes using LQG and regard inhomogeneitys as small deviations propagating on the resulting homogeneous LQC *quantum* geometries" gives a correct qualitative description of the BKL scenario but then misapplies it by referring to entire homogeneous models rather than microscopic homogeneous regions.

- "Returning to the more restricted setting of cosmology, it seems fair to say that LQC provides a coherent and conceptually complete paradigm that is free of the difficulties associated with the big-bang and big-crunch. Therefore, the field is now sufficiently mature to address observational issues" is premature, given the serious problems of the approach reviewed in [1]. In particular, quantization ambiguities, effective theory, and covariance must be under control for reliable observational predictions.

To summarize, defining assumptions made by the Ashtekar school have violated basic concepts of effective field theory, which led it to overestimate the robustness of their models and to downplay quantization ambiguities. It oversimplified the approach to high curvature, leading to unreliable statements about the quantum nature of the big bang. Even if a valid effective theory could be derived from loop quantum gravity, it would not be expected to confirm results by the Ashtekar school because an effective theory rarely makes use directly of fundamental parameters, as assumed by the Ashtekar school, and crucially includes running parameters. Morever, models developed by the Ashtekar school cannot be covariant, as shown by a no-go theorem based on the discussion from Section 4. Violating such an important consistency condition is not an approximation that may be improved by including further corrections. Models that violate essential symmetries rather produce spurious solutions which invalidate any analysis; see for instance [91] in a different context. In loop quantum cosmology, this outcome is shown by the possibility of signature change at high density in non-classical space-time structures compatible with holonomy modifications [92–94]. The transition through high density is then no longer deterministic [95,96], a result which cannot be obtained from deterministic bounce claims by adding small corrections.

We have reached a rather sobering end. It may be possible that some readers of this critique do not agree with all the conclusions drawn here. However, it should have become clear that the present status of loop quantum cosmology can accommodate a wide variety of interpretations, not just the overly optimistic view espoused by the Ashtekar school [1]. At present, it is not possible to derive a sufficiently complete effective theory from full loop quantum gravity, and even if this were possible, it is not clear whether loop quantum gravity itself is covariant and consistent. Nevertheless, as sketched in the present contribution, one can test the validity of proposed models based on general requirements on effective theories, and independently perform tests of covariance. Given presently available models, the precise behavior at Planckian densities remains undetermined, and even qualitative features are too ambiguous and uncertain to select a specific scenario. However, careful statements about the approach to high curvature may be made, indicating what implications the first deviations from classical behavior of gravity or space-time might imply.

Bringing attention to the possibility of different outcomes in light of ambiguities and uncertainties may at some point indicate a certain universal behavior, which would be missed if one insists on a presupposed outcome. Such questions may be more modest than the grand claims presented in [1], but they are well worth pursuing.

**Funding:** This research was funded by the NSF, grant number PHY-1912168.

**Conflicts of Interest:** The author declares no conflict of interest.

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
