# Peer review of "Critical Evaluation of Common Claims in Loop Quantum Cosmology"

_universe, doi:10.3390/universe6030036_

Round 1
Reviewer 1 Report
This manuscript presents is a detailed and constructive criticism of the paper [1] "Loop Quantum Cosmology: a Status Report" by Ashtekar and Singh, published by Classical and Quantum Gravity and available online as arXiv:1108.0893. Through this review of that paper, the author actually reviews, evaluates and challenges some of the standard views on loop quantum cosmology. Besides suggesting to improve some points in the presentation and to moderate some of the claims, I definitely recommend this paper for publication.
Here are points that could be improved before publication:
1. The author focuses on [1], which date back to 2011. Hasn't there been any progress since then on the various issues at hand? Isn't the author possible discarding more recent relevant work by those authors and other researchers working along the lines of the "Ashtekar school"? 2. The paper is written in a rather heavy style, with multiple quotes from the original paper [1]. Could the author clearly identify the various issues with that work, in the introduction, possibly as taglines, possibly as a bullet list, such as 'need to adjust the fidcuial cell to the scale of the inhomogeneities' (or 'take into account the feedback of the inhomogeneities at the quantum level'), 'properly working out the coarse-graining of LQC as an effective field theory with running couplings', 'lack of covariance of the formalism and proposed modification to the Einstein equations' and so on. 3. Besides the work [1], do the issues discussed by the author apply beyond LQC to the more recent application of those standard ideas to black hole models in LQG, as proposed for instance by Rovelli and collaborators, where the new phenomenology is based on a new repulsive force as in LQC. 4. In his effort towards a critical evaluation of [1], the author could be more precise and moderate in some of the claims made in the present manuscript. For instance, in the last section, when the author mentions that "models developped by the Ashtekar school can not be covariant", is there an actual no-go theorem, or should this claim be slightly modified. 5. Could the atuhor briefly summarizes which directions he believes should be investigated in LQC based on his evaluation of the standard understanding of LQC? Can he envision ways to overcome those issues?
Other minor points are: a. could the author mention in the abstract some of the signature claims made by the Ashtekar school? b.What is Theta_Lambda in the 2nd paragraph of the introduction? c. In section 2.1, what is the dimension of mu? is it a regularization parameter, supposedly at the Planck scale or resulting from another mechanism occuring at some scale between the Planck scale and the cosmological scale?
Author Response
I have implemented all these useful suggestions:
1. Much recent work has built on [1] but it always accepted the original assumptions and claims. I have inserted "While the review [1] is by now rather old, it has been foundational and is still widely believed to be valid in the
very recent literature. Most of the latter did not question the claims made in [1] but rather built on them. A detailed analysis of [1] therefore remains timely." in lines 52-55.
2. I now mention the main issues in the abstract ("such as the genericness of bounces with astonishingly semiclassical dynamics, robustness with respect to quantization ambiguities, the realization of covariance, and the
relevance of certain technical results for potential observations") and have appended a final paragraph to the introduction: "We briefly summarize the main problems to be discussed: (i) The Ashtekar school bases its constructions on the erroneous assumption that large comoving regions may be assumed to be homogeneous in the early universe. As a justification, an appeal is made to the Belinskii--Khalatnikov--Lifshitz (BKL) scenario [BKL] without noting that this scenario is valid only asymptotically close to a singularity and does not provide a generic lower bound for the size of homogeneous regions. As we will show in Section 2, this mistake implies that quantum effects and their associated quantization ambiguities are severely underestimated. (ii) The Ashtekar school implicitly makes strong assumptions about the availability of effective descriptions, thereby rendering any results based on them non-generic. In particular, the Ashtekar school assumes that a single effective theory can be used throughout a wide range of vastly different scales, including the late and the very early universe (and even through a bounce). In addition, the school assumes that a certain parameter calculated in a fundamental theory (the so-called ``area gap'') can directly be inserted in the equations of an effective theory even though no derivation of the latter from the former
theory exists at present. These problems will be discussed in Section 3. (iii) As shown in Section 4, the Ashtekar
school has made several incorrect claims about the realization of general covariance in its models."
3. They do. I have inserted the sentence "They also apply to various black-hole models that are based on replacing the singularity with a bouncing interior within the horizon." on lines 91/92, but prefer not to go into more detail to keep the review focused on cosmology (in accordance with a remark made by Referee 3 about Section 4).
4. I did try to be moderate, but also realize that the problems are rather severe and should be pointed out clearly. The arguments given in Section 4 can indeed be worked into a no-go theorem about covariance with holonomy
modifications. While I did not include a formal theorem in this paper, I have reworked and slightly extended Section 4. In particular, I now state explicitly that "The observations of [11] then rule out holonomy modifications as ingredients of slicing-independent space-time theories. They constitute a no-go theorem for the possibility of covariant holonomy modifications." in lines 652-654.
5. I had described such directions in the original submission, for instance in my sketch of numerical loop quantum cosmology in Section 3.2. I have slightly extended the final conclusions on page 20 to indicate further directions.
I now mention the main issues in the abstract, and also changed "signature claims" to "broader claims" which is perhaps more clear. I now explain that Theta_Lambda, in the notation of [1], refers to a generator of internal-time
evolution. In the paragraph following equation (5), I give further properties of mu, in particular its length units and its nature of a modifier (but not a regulator that would eventually be removed).
Reviewer 2 Report
In this article the author examines a number of assumptions and results used in LQC models, especially those pioneered by the Ashtekar school, in part in an effort to establish a more robust and rigorous foundation for LQC within the context of LQG. As indicated in line 582, at the outset efforts are made to establish any possible framework to initially make progress in a new area and it has been nearly ten years since a fairly widely regarded review of the field has appeared. To be sure, it may be quite some time before a well regraded and unique theory emerges for Planck scale spacetime models, however, it is possible to examine current progress during the last decade, especially with regard to holonomy modifications and covariance but general treatments are lacking. This article represents a summary of these shortcomings with some brief counter examples and directions for developing an improved approach. This article is direct and strongly worded, and I am not sure that “bounceology” is a good word to use, others have looked at the shortcomings of LQC, i.e. note the work of Barrau in
Barrau, Aurélien, and Boris Bolliet. "Some conceptual issues in loop quantum cosmology." International Journal of Modern Physics D 25, no. 08 (2016): 1642008,
and
Barrau, A., T. Cailleteau, J. Grain, and Jakub Mielczarek. "Observational issues in loop quantum cosmology." Classical and Quantum Gravity 31, no. 5 (2014): 053001.
While there is not universal agreement on the impact and overall intractability of these issues this is certainly one way of bringing them to light once again especially for new researchers in the field. Please consider the following issues:
1.Note in Line 1 the word "has" should be "have."
2. The narrative is clear and well written, although it includes a number of detailed quotations from other works, but the use of bounceology in line 252 might be more about singularity free cosmolog.
3. Line 343 is more indicative of the failure of effective field theor.
4. Line 454 might focus on LQC corrective effort.
5. Barrau is referenced for the anomaly free perturbation work in line 778 but you might also consider the efforts in his conceptual issues with LQC work as well.
Author Response
I thank the referee for these suggestions. I have included the two references in line 56, where I also describe the nature of issues pointed out here, which are conflicts between certain assumptions made in loop quantum cosmology and general physics principles. The issues discussed in the two references are more specific to loop quantum cosmology within its own framework.
In the first paragraph of section 2.2.2 I now discuss in which sense I am using the new term "bounceology" which refers to studies of a pre-supposed bounce (explicitly or implicitly) rather than an unbiased investigation of
possible outcomes.
By "ineffective theory" (section 3.1) I refer to an unjustified application of effective theory. This could be a wrong application of general principles of effective theory, or an extension of effective theory beyond its regime of validity, which would be a failure as indicated by the referee.
I have changed the heading of section 3.2, metaphorically referring to an improved version.
Reviewer 3 Report
The article "Critical evaluation of common claims in loop quantum cosmology" presents an overview of some issues that are present in the field of loop quantum cosmology. While not containing major new results, the article serves as a summary of some problems that have been detected over the last years. Given the state of the art in the field, highlting these issues and presenting avenues t oovercome them is very important. Overall, I therefore highly support publication of this article.
However, I have nontheless the feeling that this article can be improved in many instances in order to attract a wider audience and achieve the desired attention.
1) The title of the atricle is fairly general, however in its present form it serves mainly as an attack to the status report [1]. One should either change the title in such a way that it mentions being a critic at [1], or -- given the fact that [1] is almost a decade old and research quickly changes -- put more emphasize on the newer literature:
There are more recent reviews (e.g. arXiv:1612.01236) which should be menioned and compared regarding updates on the "common claims" as well as other works still relying on such claims.
On the same line, the fairly general title of article could be justified if further issues of LQC are mentioned. At present, the author only presents the issues on which he had worked himself over many years, neglecting contribution of other people in the field. For example, arXiv:1906.07554 presents problems regarding the dynamics in LQC and deserves mentioning.
2) The overall tone in the article is very one-sided. It could be improved towards a more neutral attitude -- especially if the LQC community is supposed to learn from the investigations, the later ones should not be perceived as an unfair attack. Albeit agreeing with the authors intention, I have the feeling that some statements are unjustified, e.g.: "None of the signature claims [...] hold upt to scrutiny" (some claims persists, such that at least certain models feature resolution of singularity) "Quantization ambiguity are downplayed by an incorrect understanding of effective theory" (especially nowadays LQC puts more emphasis on quantisation ambiguities since arXiv:1706.09833 and follow up studies) "The Ashtekar school does not solve the problem but rather evades it..." (to my understanding the observable framework is a perfectly valid way to make diffeomoprhism-invariant statements -- if deparametrisation has an intrisic problem the author should state it. If referring to the choice of added matter, please see that active research goes into studies of different forms of matter as well, e.g. arXiv:1108.1145) "Bouncing solutions are guaranteed, but only by an implicit sleight of hand" (According to my understanding the mentioned sleight here is not the reason for the bounce. The later happens not due to these quantum properties but is also featured in the (doubtable) classical effective setting and thus rather due to the modified dynamics.) ... There are many more of these statements which I believe are "too strong" and could be soften appropriately in order to ensure that the article is perceived to be scientifically neutral and fair.
3) Some minor technical points :
-[p3, after (4)] "The next step introduces modifications by a discrete structure of space in loop quantum gravity" -- LQG is NOT a discrete theory of Quantum Gravity. It design aims at constructing a continuous QFT and the spinnetwoks over (all!) finite graphs form only a basis. Continuous geometry is suppsed to arise by suitbale summation or limit techniques. This is independent from the statement that "eigenvalues of certain operators have discrete spectrum" and should carefully be kept in mind at various instances over the article
-[p4.101 & 103] Small typos: "unsing homogenous a model" and "cingularity"
-[p.5 106] Missing citation
-[p. 11. 349 and others] "derived from a fundamental theory (which at present is not possible...". One should nonetheless mention that many promising approach towards such a derivation are on its way, e.g. the QRLG programm for coherent states and constructions of the cosmological subsector in LQG arXiv: 1706.02424 and arXiv:1911.09639
-[p14 ff] Section "Covariance". The article shifts its focus a bit too much in this section from LQC to black holes. For cosmology experts not profound in BH it might be hard to understand how [7] can be applied in cosmology. I would prefer an extension of this discussion before a claim as "deformations of space-time structure [...] invalidate existing attempts to derive parameters of an effective theory of loop quantumcosmology" is made.
Author Response
I have made the suggested changes or included further explanations of some statements commented on by the referee.
1) In my opinion, the title is specific enough because it refers to "common claims" which does not include technical issues as pointed out for instance in arXiv:1906.07554. (Nevertheless, I have included this reference as well as a
related more recent one.) The specific issues discussed in my paper are now mentioned explicitly in the abstract. Regarding the age of [1], its main assumptions are still being used in much of the current literature. The more
recent review arXiv:1612.01236, for instance, merely provides an update on [1] but does not question in any way its principles. I now describe this situation in lines 52-60: "While the review [1] is by now rather old, it has been
foundational and is still widely believed to be valid in the very recent literature. Most of the latter did not question the claims made in [1] but rather built on them. A detailed analysis of [1] therefore remains timely. Other critical viewpoints have occasionally been presented, such as [two references suggsted by Referee 2] which discuss observational questions and the robustness of bounces but do so within the setting described in [1]. Others, such as [arXiv:1906.07554 and a more recent related reference], focus on technical questions of a specific Hilbert-space representation. Our discussion here will be broader (and therefore much more damning), pointing out several violations of general physics principles such as the ubiquity of quantization ambiguities, the domain of effective field theory, and the condition of general covariance."
2) I do believe that my statements are justified. Because they describe severe problems in the framework set up in [1], they have to be made as clearly as possible. While it is true that some technical claims in [1] are correct, such
as certain solutions in specific models, a model is only as good as the assumptions used to construct it. If the assumptions are wrong, the model or claims extracted from it are not of much use. In this sense, I refer to
"broader claims" (previously, "signature claims") in the abstract. I also mention the specific broader claims in the abstract.
I have inserted a statement about ambiguities studied in loop quantum cosmology, see lines 220-226. However, these are ambiguities in possible phenomenological scenarios of bouncing models. They do not consider the possibility that ambiguities might even challenge the presence of a bounce, which is a more fundamental question on which I focus in my paper.
I have explained the problem with deparameterization in somewhat more detail at the beginning of section 2.2.1. The problem is not that a specific choice of matter is usually made to implement the procedure, which can certainly be
varied as indicated by the referee. In these cases, one would consider different physical models for the nature of an internal time. The problem is that in models in which multiple choices of a global internal time are possible, for instance if one includes a free massless scalar as well as dust in the same model, different choices generically give rise to inequivalent quantum corrections. Therefore, deparameterization does not imply (time diffeomorphism invariance at the quantum level.
At the end of section 2.2.1 I now recall briefly that two broader assumptions are required for generic bounces: (i) holonomy-modified dynamics and (ii) a specific kind of dynamical representation. The former leads to an sl(2,R)
structure, which would still be compatible with non-bouncing solutions in some of its representations. Choosing a specific representation (from the positive discrete series) is the "implicit sleight of hand" I referred to. I have gone
through the entire manuscript and either explained critical statements in more detail or weakened them.
3) The intended meaning was not that loop quantum gravity discretizes space by restricting continuous fields to lattices, although I agree that it could have been misunderstood. I have expanded the statement by saying that "While loop quantum gravity is a quantum theory of continuum fields, it leads to discrete spectra of geometrical operators such as the spatial volume. The corresponding quantum representation implies that the Hamiltonian constraint cannot be quantized directly in its classical form and has to be modified. As we will indicate briefly in Section 4, the brackets of constraints then carry information about a modified space-time structure, indirectly related to the discreteness of geometrical spectra. The modifications to be discussed now, initially motivated by formal properties of a quantum representation, are therefore indicative of potential dynamical effects implied by discrete
space." in lines 135-142.
I have corrected the typos (as well as a few others). The question mark was not a missing citation but rather an indication that the reference to Occam's razor was not quite right. I have removed the question mark and added a note in the parenthesis that follows the quote.
I do not think that the QRLG program or other definitions of cosmological sectors are helpful in the given context, and discussing these approaches in the paper would be too much of a digression. In brief, in these approaches it is not clear how one can meaningfully implement a coarse-graining procedure of solutions of the full theory (and not just reductions of some operators). Instead of these directions, I have referred to various approaches to spin foam renormalization at the bottom of page 15.
I have rearranged the discussion in Section 4, drawing a new parallel with LTB models instead of just considering black hole exteriors. I have also expanded the bottom paragraph on page 17, indicating how these considerations can be worked into a no-go theorem for covariant holonomy-modifications.
Round 2
Reviewer 3 Report
The Author has addressed all my previous comments and suggestions. I believe that the article is well-suited for publication